# Evaluation of expression variations in virulence-related genes of *Leishmania major* after several culture passages compared with *Phlebotomus papatasi* isolated promastigotes

**Taher Nemati Haravani[1], Parviz Parvizi[2], Seyed Hossein Hejazi[1], Mohammad Mehdi Sedaghat[3], Abbasali Eskandarian[1]\*, Mahmoud Nateghi Rostami[4]\***

**1** Department of Parasitology and Mycology, School of Medicine, Isfahan University of Medical Sciences, Isfahan, Iran, **2** Molecular Systematics Laboratory, Parasitology Department, Pasteur Institute of Iran, Tehran, Iran, **3** Department of Medical Parasitology and Mycology, School of Public Health, Tehran University of Medical Sciences, Tehran, Iran, **4** Parasitology Department, Pasteur Institute of Iran, Tehran, Iran

\* dreskandarianstudents2@gmail.com (AE); M_Nateghi@Pasteur.ac.ir (MNR)

**Data Availability Statement:** All relevant data are within the paper.

## Abstract

Cutaneous leishmaniasis (CL) is a prevalent infectious disease with considerable morbidity annually. Here, we aimed to investigate the likely variations in gene expression of glycoprotein63 (gp63), heat shock protein 70 (HSP70), histone, arginase, cysteine protease B (CPB), *Leishmania* homologue of receptors for activated C kinase (LACK), small hydrophilic endoplasmic reticulum-associated protein (SHERP) in metacyclic promastigotes of *L. major* isolated from *Phlebotomus papatasi* sand flies and promastigotes excessively cultured in culture medium. The parasites were collected from suspected CL cases in Pasteur Institute of Iran, cultured and inoculated into the female BALB/c mice ($2 \times 10^6$ promastigotes). Sand flies were trapped in Qom province, fed with the blood of euthanized infected mice and subsequently dissected in order to isolate the midgut including stomodeal valve. The metacyclic promastigotes were isolated from *Ph. papatasi* (Pro-Ppap) using peanut agglutinin test (PNA), then continuously cultured in RPMI-1640 medium enriched with fetal bovine serum, penicillin (100 U/ml) and streptomycin (100 mg/ml) to reach stationary phase (Pro-Stat). The gene expression was evaluated in both parasitic stages (Pro-Ppap and Pro-Stat) using qRT-PCR. Out results showed a significant increased gene expression at Pro-Ppap stage for gp63 ($P = 0.002$), SHERP ($P = 0.001$) and histone ($P = 0.026$) genes, in comparison with Pro-Stat stage. Noticeably, significant changes were, also, demonstrated in 10th to 15th passages [gp63 ($P = 0.041$), arginase ($P = 0.016$), LACK ($P = 0.025$)] and in 5th to 20th passage (SHERP) ($P = 0.029$). In conclusion, the findings of the present study seem to be essential in designing *Leishmania* studies, in particular regarding host-parasite interaction, immunization and infectivity studies.

**Funding:** The author(s) received no specific funding for this work.

**Competing interests:** The authors have declared that no competing interests exist.

## 1. Introduction

Leishmaniases, are caused by infection to some species of genus *Leishmania*, a parasitic flagellated protozoon that are neglected vector-borne tropical diseases with diverse clinical outcomes. They being spread throughout old world and new world territories [1]. The female of the *Phlebotomus* and *Lutzomyia* genera (Diptera: Psychodidae) are the vectors for parasite transmission [2]. Three types of disease may occur, including cutaneous leishmaniasis (CL), visceral leishmaniasis (VL) and muco-cutaneous leishmaniasis (MCL), among which VL is the deadliest and CL is the most prevalent [3]. The annual incidence rate of CL has been estimated to be 700,000 to 1.2 million cases, mostly reported from Brazil, Colombia, Algeria, Syrian Arab Republic, Islamic Republic of Iran and Afghanistan [4]. In Iran CL manifests as two common forms: **i**) anthroponotic cutaneous leishmaniasis (ACL) or dry form, caused by *Leishmania tropica* (*L. tropica*) and human and dogs act as reservoir hosts, **ii**) zoonotic cutaneous leishmaniasis (ZCL) or wet form, due to *L. major* and rodent reservoir hosts [5]. *Phlebotomus papatasi* and *Ph. sergenti* are considered as major vectors for ZCL and ACL, respectively [6]. Long-time treatment course, side-effects of drugs and drug resistant issue are the major concerns regarding leishmaniasis [7]. Virulence factors (VF) in *Leishmania* parasites are key components in pathogenesis, among which leishmanolysin (gp63) [8], Heat shock protein 70 (HSP70) [9], Arginase [10], Cysteine proteases (CP) [11] and *Leishmania* activated C kinase (LACK) [12] are significantly involved. Thus far, such VF have been interested as eminent drug and vaccine targets against leishmaniasis.

Upon sand fly blood feeding from infected vertebrate host, the amastigotes give rise to the slowly-dividing procyclic promastigotes within midgut. In the following, the latter differentiate into nectomonad, then efficiently-dividing leptomonad promastigotes, while the parasites approach the thoracic midgut. Next, highly-infectious non-dividing metacyclic promastigotes would develop from leptomonad forms, which can be readily injected to a new host and establish the infection [13]. At the vertebrate host interface, the inoculated metacyclic promastigotes are phagocytized by the circulatory and/or resident dermal macrophages and carried to the lymph nodes or lymphatic organs in case of CL and VL, respectively. About a week after sand fly bite, primary local immune responses are relieved and an early lesion gradually forms at the biting site [14].

In addition to the isolation from sand flies, promastigotes can be harvested and purely cultivated in axenic culture media in large volumes [15]. Nevertheless, it has been reported that long-lasting cultures may result in decreased virulence and increased drug tolerance in *Leishmania* promastigotes [16]. Certainly, temperature and pH differences between sand fly and continuous cultures are highly influential in gene expression variations and consequently parasite virulence. Based on previous literature, an increased temperature up to 37 ˚C or declined pH could substantially induce parasitic transformation from promastigotes to amastigotes and elicit gene expression and virulence changes [17–19]. A study on *L. infantum* promastigotes showed stage-specific differential gene expression between excessive cultures and parasites isolated from *Ph. perniciosus* foregut. For example, downregulation of glucose-6-phosphate N-acetyltransferase (GNAT) gene was observed in amastigotes in comparison with cultured stationary and logarithmic phase promastigotes and metacyclic promastigotes in sand fly. Similar observations were, also, found regarding the sodium stibogluconate resistance protein (SbGRP) and α-tubulin genes [20]. Altogether, realization of protein expression patterns and metabolic differences in metacyclic promastigotes originated from sand fly foregut or axenic cultures and amastigotes assists us to discover novel molecular targets with specific implication for more efficacious drug/vaccine design and the production of specific monoclonal antibodies.

Among vital *Leishmania* genes important for survival, most studies have focused on seven VF genes, including gp63, heat shock protein 70 (HSP70), Arginase, cysteine protease B (CPB), *Leishmania* homologue of receptors for activated C kinase (LACK), small hydrophilic endoplasmic reticulum-associated protein (SHERP) and histone; thus the present study was done on these VF-related genes in *L. major* promastigotes isolated from excessive cultures compared with those isolated from *Ph. papatasi* sand flies in order to shed light on the comparative expression patterns of these genes in the sand fly microenvironment and in the continuous cultures and translate the findings to the biological activity of the parasite.

## 2. Materials and methods

### 2.1. Parasite collection and maintenance

In this study, four suspected CL patients referring to the Pasteur Institute of Iran were sampled under aseptic conditions and referred to the Parasitology section of the Pasteur Institute of Iran for further identification. The parasites were initially cultured in Novy-MacNeal-Nicolle (N.N.N.) medium, then inoculated into female BALB/c mice ($2\times10^6$ promastigotes). Four weeks later, mice were euthanized using a $CO_2$ chamber, dissected and amastigotes were cultured in Roswell Park Memorial Institute (RPMI 1640) medium (pH: 7.2), supplemented with 10–15% inactivated fetal bovine serum (FBS), Penicillin (100 U/ml) and streptomycin (100 mg/ml) and incubated at 24 ˚C. Consecutive passages were prepared of four clinical specimens and logarithmic and stationary phase promastigotes were pelleted upon triple washing with phosphate buffered saline (PBS) and kept at -20 ˚C for further use.

### 2.2. Collecting, infecting and dissecting *Ph. papatasi* sand flies

Due to the proximity to Tehran province, those CL-endemic areas in Qom province were selected for sand fly collection, which could be transferred to Tehran at the earliest time. Qom city is the 7[th] largest metropolis in Iran, located at 140 km south of Tehran.

Live sand flies were collected using CDC miniature light trap, funnel trap and aspirator. Collected sand flies were transferred to the Pasteur Institute of Iran using steel cages (35×35×35 cm) under insectarium conditions. Engorged, pregnant, male and female sand flies were separately sorted and maintained in polyester cages (5.5 cm height, 4 cm diameter). Female BABL/c mice infected with *L. major* metacyclic promastigotes three months ago were anesthetized by ketamine hydrochloride (60 mg/kg) and xylazine (15 mg/kg) and exposed for 1 h to sand fly bites (starved up to 24 h). The blood fed sand flies were maintained in insectarium for an additional 5 days under optimum conditions (27–28˚C, 90–100% relative humidity, 17 h light—7 h darkness photoperiod), their midgut (including stomodeal valve) were totally dissected in PBS solution under stereo microscope and the anterior part of thoracic midgut stomodeal valve and *Ph. papatasi* promastigotes (Pro-Ppap) was isolated. The Pro-Ppap in a PBS drop were harvested from the midgut upon a slight pressure using a cover slip. Each Pro-Ppap sample contained material from 20 sand flies.

### 2.3. Isolation of metacyclic promastigotes from sand flies and culturing

The peanut agglutinin test (PNA) was employed to isolate the *L. major* metacyclic promastigotes. In this method, a specific lectin binds to the galactose of lipophosphoglycan (LPG) molecule, that is only formed at the surface of metacyclic promastigotes, causing the differentiation and isolation of these parasitic stages. In brief, $2\times10^8$ parasites/mL were seeded in 96-well plates containing PNA media and 50 μg/mL peanut agglutinin was used along with Hanks balanced salt solution (HBSS), without $NaHCO_3$, for parasite suspension at ambient temperature

for 30 min. PNA-agglutinated or unagglutinated promastigotes were triple subjected to HBSS to eliminate suspended particles. In addition, the promastigotes $2 \times 10^8$ to $5 \times 10^8$ were simultaneously suspended in HBSS with 100 µg/ml of PNA. Upon incubation for 30 min, the parasites were centrifuged at 2000×g for 5 min. Ultimately, agglutinated parasites were twice washed and counted.

The isolated metacyclic forms were initially passaged three times in N.N.N. medium, under aseptic conditions, and they were checked daily regarding parasite growth and likely contaminants. In case of progressive parasite growth, they were transferred to the RPMI 1640 medium for mass production. Briefly, a volume of N.N.N. supernatant containing grown parasites were transferred to the RPMI 1640, enriched with 10–20% FBS, Penicillin (100 U/ml) and streptomycin (100 mg/ml). The culture flasks were incubated at 25 ˚C. Once a considerable rate of parasites was grown and were at the metacyclic (stationary) phase (Pro-Stat), enriched culture medium was continuously added.

After mass production, consecutive culture passages were prepared under aseptic conditions, and were monitored at constant temperature and pH conditions. In order to spot virulence-associated gene expressions in consecutive passages of *L. major* (1, 5, 10, 15 and 20), the metacyclic promastigotes were isolated from consecutive cultures using PNA method, as mentioned earlier.

## 2.4. RNA isolation and complementary DNA (cDNA) synthesis

Total RNA extraction from consecutive passage of metacyclic promastigotes from cultures and sand flies was done using TRIzol reagent, according to the manufacturer's protocol. Briefly, this solution was added to the pelleted promastigotes 500 mL/$10^6$ cells and incubated in room temperature for 10 min. Then, 200 mL chloroform was added, the mixture was vortexed (2 min) and incubated for 15–20 min. After centrifugation (12000×g, 15 min, 4 ˚C), the supernatant containing RNA was transferred to a DNase-RNase free micro tube and after addition of 500 mL isopropanol, 15 min incubation was done on ice. Again, centrifugation was performed (12000×g, 10 min, 4), the supernatant was discarded and 100 mL cold ethanol 75% was added and vortexed. A final centrifugation was done at 12000×g, 5 min, 4 ˚C, the supernatant was removed and 30 mL pre-warmed DEPC RNase-free water was added to the pellet. The concentration of the extracted total RNA was measured using a NanoDrop device and were kept at -80 ˚C for further use.

In the following, cDNA synthesis was performed based on the kit protocol. Briefly, purified RNA samples were placed on ice and a total reaction volume of 20 µl was prepared in a streel micro tube, comprising 8 mL purified RNA, 10 mL mix buffer, 2 mL enzyme. The experimental steps were including consecutive incubation at 25 ˚C, 47Ċ and 85˚C for 10, 60 and 85 min in a water bath, followed by incubation on ice and transferring to the -20 ˚C.

## 2.5. Quantitative real-time reverse transcriptase polymerase chain reaction (qRT-PCR)

Relative quantification was done to determine the variations in gene expression using $2^{-\Delta\Delta ct}$ method and Glyceraldehyde-3-phosphate dehydrogenase (GAPDH) was used as internal control or house-keeping gene. Details of used primer pairs in this study are provided in Table 1. The final mixture (20 mL) included 10 mL of 2X qPCRBIO SYGREEN Hi-ROX (London, UK), 0.5 mL of each forward and reverse primers, 1.5 mL synthesized cDNA as template and 7.5 mL distilled water. The amplification program was as follows: Initial denaturing step at 95 ˚C (5 min), followed by 35 cycles of extension at 95 ˚C (30 sec), annealing at 60 ˚C (1 min) and

**Table 1. The primers used for the amplification of seven virulence genes in *L. major* and GAPDH gene (RT- PCR).**

| Gene name | | Primer Sequence | Product size (bp) | References |
|---|---|---|---|---|
| GP63 | F | TCATGGACTACTGCCCTGTC | 156 | [52] |
| | R | CTATGCCGTCAGTTGCCTTC | | |
| HSP70 | F | GCAACCAGATCACCATCACC | 150 | [53] |
| | R | AGTACGCGTAGTTCTCCAGG | | |
| Arginase | F | TCC CGA GTG CTT TTC GTG G | 138 | [10] |
| | R | TCC ACG TGA TGC ATG CTG AA | | |
| CPB | F | ACAGCTCCT−CTTTCATGGAC | 92 | [54] |
| | R | AATGTGTGAGGA−CAGGTACG | | |
| LACK | F | GAACTACGAGGGTCACCTGAA | 165 | [55] |
| | R | AGACCGTAGTCGCTGTCCAC | | |
| SHERP | F | CAATGCGCACAACAAGATCCAG | 120 | [56] |
| | R | TACGAGCCGCCGCTTATCTTGTC | | |
| Histone | R | ACACCGAGTATGCG | 85 | [56] |
| | F | TAGCCGTAGAGGATG | | |
| GAPDH | F | AGACCGTAGTCGCTGTCCAC | 226 | [57] |
| | R | GAAGATGGTGATGGGATTTC | | |

72 ˚C (10 sec). A final extension step was also performed at 95 ˚C for 15 sec. Ultimately, the temperature was decreased to 60 ˚C for 1 min.

## 2.6. Statistical analysis

The one-way ANOVA and the Posthock tests were used for comparing the quantitative values between and in the different groups. The confidence level was considered 95% ($\alpha \leq 0.05$).

The GraphPad Prism and IBM SPSS v.25 computer programs were used for drawing the graphs and analyzing the obtained data respectively.

## 3. Results

The qRT-PCR data for gp63 gene, demonstrated that this gene undergoes highly significant variations in expression rate in the sand fly, in comparison with consecutive culture passages ($P = 0.002$). A significant increase in gp63 gene expression between the 10th to 15th passages (P10 to P15) observed ($P = 0.041$), while the expression between other passages did not reach to significant level ($P = 0.05$) (Fig 1A). Regarding HSP70 gene, no meaningful variation was demonstrated between sand fly and various culture passages parasites ($P = 0.237$) (Fig 1B). Insignificant expression patterns were, also, demonstrated in arginase ($P = 0.082$), CPB ($P = 0.082$) and LACK ($P = 0.237$) genes (Fig 1C, 1D and 1F). With respect to Pro-Stat, there observed a statistically remarkable increased expression between P10 and P15 for arginase ($P = 0.016$) and LACK ($P = 0.025$).

Another prominent finding of the present study was a dramatic increase in expression of SHERP ($P = 0.001$) and histone ($P = 0.026$) genes in the vector, comparable to the consecutive culture passages promastigotes (Fig 1E and 1G). It is, also, noteworthy that a substantial SHERP expression was demonstrated between P5 to P20 passages, which was statistically significant ($P = 0.029$). Altogether, among 7 examined genes in the present study, 3 genes (gp63, SHERP and histone) showed significantly higher expression in Pro-Ppap than in Pro-Stat in successive passages.

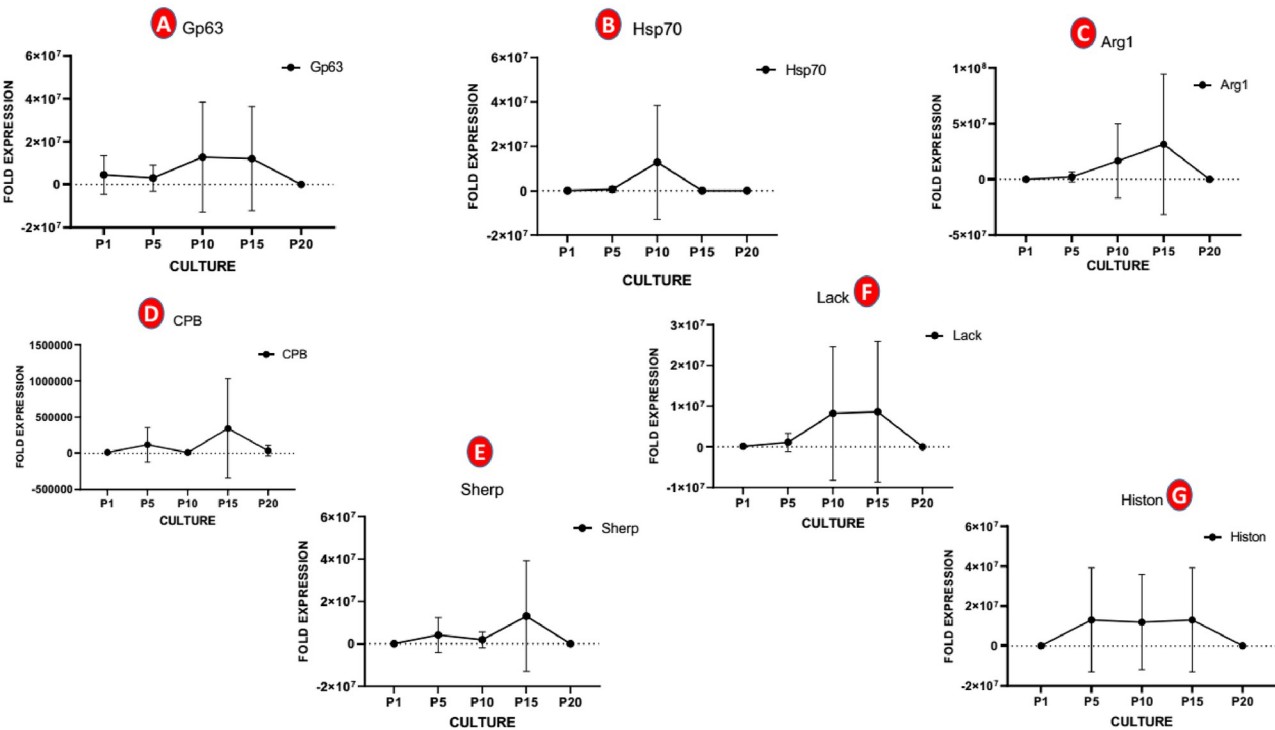

**Fig 1. Differential expression pattern analysis of seven VF-related genes (gp63, HSP70, arginase, CPB, LACK, SHERP and histone) in *L. major* Pro-Stat and Pro-Ppap stages, using quantitative real-time PCR.**

## 4. Discussion

Since studying *Leishmania* promastigotes is struggling in their natural milieu *i.e.* the sand fly gut, due to the reduced biomass and manipulation, axenic parasite cultures in liquid media were evolved in the 1960s and 70s, in order to simulate the sand fly gut conditions for *in vitro* promastigote culture [21–24]. The promastigotes can be readily cultivated in such media enriched with trace elements, lipoid substances, proteins and low molecular weight nutrients. Over a week, cultured promastigotes pass stationery and death phases, respectively; however, a noticeable quantity of promastigotes remain alive for weeks [25, 26]. Interestingly, upon multiple culture passages, some original features, virulence and infectivity of promastigotes may attenuate, hence a passage through laboratory animals may be demanded. This finding may substantially influence the outcome of studies on *Leishmania* infectivity, immunization and host-parasite interaction [20].

Trypanosomatids protozoa such as *Leishmania* have a complex life cycle, involving invertebrate and vertebrate hosts, as well as alternating cycles of morphological and infectivity variations [27]. Stage-specific gene regulation is an extremely critical step in such processes. On this account, elucidation of the expression profile of particular genes provides insights into the intricate biology of *Leishmania* parasites [28]. In this sense, it was shown that intracellular *L. mexicana* [29] and *L. infantum* [25, 26] amastigotes differ from axenically cultured amastigotes in terms of expression profiling. Another astonishing finding was that a plethora of transient or permanent variations in gene expression occur while differentiation of *L. donovani* promastigotes to amastigotes [30]. Alcolea *et al.* (2009) reported that those genes directly or indirectly associate with infectivity are upregulated in metacyclic PNA⁻ promastigotes of *L. infantum* [31]. In the following, Alcolea *et al.* (2010) demonstrated that during promastigote

to amastigote differentiation, acidification and mostly temperature increase are critical biological factors [25]. Over two decades ago, initial attempts to decipher the differential expression of *L. major* genes was done on β-tubulin during metacyclogenesis process (1996) [32]. Years later, Rochette *et al.* (2008) demonstrated the gene expression profiling variations among *L. major* and *L. infantum* [33], and Dillon *et al.* (2015) showed expression profiles between *L. major* life stages and in murine macrophage host cell [34]. Inbar *et al.* (2017) compared the midgut stages of *L. major* with the mammalian amastigote stages, and showed a series of biological activities, leading to parasite differentiation [35]. Proteomics-based analyses on the stomodeal valve promastigotes are currently difficult to perform, thus mRNA amplification is an alternative procedure to overcome such limitation [20]. Bearing this in mind, we evaluated the expression profile of these VF-related genes in *L. major* promastigotes isolated from *Ph. papatasi* and in several culture passages.

Here, we found significantly increased expression in gp63 gene in the sand fly stages (Pro-Ppap) than in consecutive culture passages (Pro-Stat), as illustrated in Fig 1A, which conforms previous studies by Inbar [35] and Alcolea [36] studies. Reportedly, seven genes in *L. major* encode the 63 kDa surface proteinase, gp63 metalloprotease (leishmanolysin), being associated with macrophage interaction, parasite survival within macrophages and complement resistance [37]. Previously, it was shown that mRNAs of various gp63 alleles were upregulated in procyclic and later developmental stages in the sand fly [35]. Traditionally, the gp63 has been correlated with metacyclic promastigotes and enhanced infectivity phenomenon [37, 38], which along with other virulent proteins (LPG, glycosylphosphatidylinositol [GPI] and proteophosphoglycans [PPG]) probably prepares the parasite for differentiation and survival within phagolysosome of macrophages, based on pre-adaptive hypothesis [39–41].

Another prominent finding of this study was a promoted expression of SHERP protein in Pro-Ppap and intermittent increased expression between P5 and P15 in Pro-Stat. This protein forms a complex with the hydrophilic acylated surface protein (HASP), encoded by LmcDNA16 locus on chromosome 23, and thought to be essential elements in metacyclogenesis step in *Ph. papatasi* vector [42, 43]. Previously, numerous SHERP mRNAs were detected in metacyclic promastigotes, whereas HASP remained upregulated also in amastigotes [35]. In line with our results, Alcolea and colleagues (2019) demonstrated that upregulated expression of HASP/SHERP gene cluster in vector-derived promastigotes than in cultured cells endorses a more successful metacyclogenesis within mosquito gut than in culture. In other words, tiny changes in the microenvironment could substantially influence the differentiation process [36]. This contrasts Alcolea *et al.* (2016) finding, where HASP/SHERP expression in the culture (Pro-Stat) was higher; the authors remarked that such increased expression in promastigotes may be a response to the absence of specific environmental markers associated with the sand fly gut microenvironment [44]. Our findings, also, are consistent with two transcriptome studies showing elevated *L. major* and *L. infantum* SHERP expression within natural vectors, *Ph. duboscqi* and *L. longipalpis* during late infection [35, 45]. Altogether SHERP is a significant metacyclogenesis marker, which may have a regulatory role in autophagy-dependent vacuolar acidification within vector host, due to its localization to *L. major* endoplasmic reticulum and mitochondrion and potential of both for autophagic digestion [46].

Histone proteins are important molecules in the cell cycle process. Previously, multiple histones (H1, H2A, H2B and H4) were shown to be downregulated during metacyclogenesis [47–49]. As the parasite approaches metacyclic promastigote stage, a declining trend in histones transcripts may be observed, as a cell cycle-dependent regulation, according to higher eukaryotes [50, 51]. Inbar showed that histones are strongly expressed during amastigote to procyclic promastigote stages, whereas the expression falls in nectomonad stage and only modestly elevates in metacyclic phase [35]. In current study, *L. major* histone expression was higher in

mosquito metacyclic stages (Pro-Ppap) than in stationary promastigotes in consecutive culture passages (Pro-Stat), as evidenced in Fig 1. Actually, the expression of histone is at a modestly higher level, in comparison with culture stages, which is consistent with aforementioned studies.

The potential limitations of the present study were including: **i**) limited funding provided by the Deputy of Research and Technology of Isfahan University of Medical Sciences, and **ii**) the simultaneous onset of the COVID-19 pandemic with the sampling sandfly season in the endemic regions, which followed by the strict quarantine rules in the country and the distribution of the viral infection. Nevertheless, to our knowledge, our study represented the significance of the expression dynamics of some VF-associated genes in *L. major* pathogenesis which can be directed towards better control of CL in the endemic regions of Iran through improved drug and/or vaccine design.

## Conclusion

As a final word, a significantly higher expression was demonstrated in three out of examined *L. major* genes (gp63, SHERP and histone) in Pro-Ppap stage, as compared with Pro-Stat phase. Nevertheless, other examined genes, including arginase, CPB, LACK and HSP70, did not show any remarkable changes between two stages. These remarkable differences in stage-regulated gene expression pattern seem to be essential for species-specific adaptations to insect or culture media microenvironments. In other words, *Leishmania* parasites may have adopted a complex and dynamic gene regulation process in response to constantly changing environments. This unprecedented variation may have a substantial impact on the host-parasite interaction and the outcome of infectivity and immunization studies. Hence, such subtle point should be emphasized in *Leishmania* studies in the future.

## Acknowledgments

The authors appreciate the staff of Parasitology Section, Pasteur Institute of Iran, Tehran, as well as Parasitology Department of Isfahan University of Medical Sciences, Isfahan, Iran. Current manuscript was issued from Ph.D. thesis of Taher Nemati.

## Ethical approval

The manuscript was ethically approved by the Ethics Committee of Isfahan University of Medical Sciences, Isfahan, Iran (Code No: 398634).

## Author Contributions

**Conceptualization:** Taher Nemati Haravani, Abbasali Eskandarian, Mahmoud Nateghi Rostami.

**Data curation:** Taher Nemati Haravani, Parviz Parvizi, Seyed Hossein Hejazi, Mohammad Mehdi Sedaghat, Abbasali Eskandarian, Mahmoud Nateghi Rostami.

**Formal analysis:** Taher Nemati Haravani, Parviz Parvizi, Mohammad Mehdi Sedaghat.

**Funding acquisition:** Parviz Parvizi, Seyed Hossein Hejazi.

**Investigation:** Taher Nemati Haravani, Mohammad Mehdi Sedaghat.

**Methodology:** Taher Nemati Haravani, Seyed Hossein Hejazi, Mohammad Mehdi Sedaghat, Abbasali Eskandarian, Mahmoud Nateghi Rostami.

**Project administration:** Seyed Hossein Hejazi, Abbasali Eskandarian, Mahmoud Nateghi Rostami.

**Resources:** Parviz Parvizi, Seyed Hossein Hejazi, Abbasali Eskandarian.

**Supervision:** Abbasali Eskandarian, Mahmoud Nateghi Rostami.

**Validation:** Taher Nemati Haravani, Abbasali Eskandarian, Mahmoud Nateghi Rostami.

**Writing – original draft:** Taher Nemati Haravani.

**Writing – review & editing:** Taher Nemati Haravani, Parviz Parvizi, Seyed Hossein Hejazi, Mohammad Mehdi Sedaghat, Abbasali Eskandarian, Mahmoud Nateghi Rostami.

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
