## [Decision Letter · Decision Letter 0]

6 Feb 2023

PONE-D-22-35287Evaluation of Expression variations in virulence-related genes of Leishmania major after several culture passages compared with Phlebotomus papatasi isolated promastigotesPLOS ONE

Dear Dr. Eskandarian, 

Thank you for submitting your manuscript to PLOS ONE. After careful consideration, we feel that it has merit but does not fully meet PLOS ONE’s publication criteria as it currently stands. Therefore, we invite you to submit a revised version of the manuscript that addresses the points raised during the review process.

We look forward to receiving your revised manuscript.

Kind regards,

Alireza Badirzadeh

Academic Editor

PLOS ONE

Journal Requirements:

2. Thank you for including the following ethics statement on the submission details page:

'The manuscript was ethically approved by the Ethics Committee of Isfahan University of Medical Sciences, Isfahan, Iran (Code No: 398634).'

Please specify if the approval includes both animal and human research in your study. In addition, please report the method of animal euthanasia in the Methods section.

Reviewers' comments:

Reviewer's Responses to Questions

**Comments to the Author**

1. Is the manuscript technically sound, and do the data support the conclusions?

Reviewer #1: Yes

Reviewer #2: Yes

2. Has the statistical analysis been performed appropriately and rigorously? 

Reviewer #1: Yes

Reviewer #2: Yes

3. Have the authors made all data underlying the findings in their manuscript fully available?

Reviewer #1: Yes

Reviewer #2: Yes

4. Is the manuscript presented in an intelligible fashion and written in standard English?

Reviewer #1: Yes

Reviewer #2: Yes

5. Review Comments to the Author

Reviewer #1: The manuscript evaluates the expression variations in virulence-related genes of L. major after several culture passages compared with wild parasites. The title sound novel and would represent valuable information on virulence-related genes that can be finally translated into the clinical settings.

The utilized methods have been appropriately applied towards the manuscript goal. There is only some typographical and stylistic errors in English writing in the manuscript, mostly in introduction and discussion parts, which should be obviated.

In total, the manuscript has been carefully devised and performed, and will be of great interest to the readers of PLoS One journal.

Reviewer #2: This manuscript presents an interesting concept and is well designed. There are some comments that must be addressed:

-Please write the abstract in a structural form. See the journal guidelines.

-Line 41-42: Please write the full form of gp63, HSP70, LACK, SHERP, CPB in the abstract first, and then write its abbreviation.

-Line 51: Please correct the phrase "Out results showed a significant increased gene..."

-Line 52-54: Please state any significant differences in the abstract text statistically and write p-value.

- According to the sentence of line 99. Explain the reason for choosing the Stationary phase compared whit the logarithmic phase.

-Please mention the reason for choosing seven genes among different virulence factors in the introduction section.

-In line 125, please write Ph. papatasi in italic form and consider in other parts of the text.

-In line 182: write the type of Real Time PCR system and Master Mix in detail.

- In line 185, it refers to the (Table), while there is no table in the text of the manuscript. Please check.

-Please mention the limitations and recommendations in the last paragraph of the discussion.

6. PLOS authors have the option to publish the peer review history of their article (what does this mean?). If published, this will include your full peer review and any attached files.

Reviewer #1: **Yes: **Dr. Hamidreza Majidiani

Reviewer #2: No

---

## [Author Response · Author response to Decision Letter 0]

16 Feb 2023

PONE-D-22-35287

Evaluation of Expression variations in virulence-related genes of Leishmania major after several culture passages compared with Phlebotomus papatasi isolated promastigotes

PLOS ONE

Dear editor and reviewers

The authors thank for spending time on this MS and giving very valuable comments; here, we have prepared the revised version and hope to find it appropriate. 

Editors comments:

2. Thank you for including the following ethics statement on the submission details page:

'The manuscript was ethically approved by the Ethics Committee of Isfahan University of Medical Sciences, Isfahan, Iran (Code No: 398634).' Please specify if the approval includes both animal and human research in your study. In addition, please report the method of animal euthanasia in the Methods section.

Response: the euthanasia method was added to the methods section. 

Response: no animal studies were performed. Regarding human samples, we only indirectly gathered those samples that had already been taken from patients by the Institute Pasteur of Iran; hence, we did not directly involve in the sampling procedure. 

4. In your Data Availability statement, you have not specified where the minimal data set underlying the results described in your manuscript can be found. 

5. PLOS requires an ORCID iD for the corresponding author in Editorial Manager on papers submitted after December 6th, 2016.

Response: the ORCID ID was matched for the corresponding author. 

Reviewers’ comments:

Reviewer #1: The manuscript evaluates the expression variations in virulence-related genes of L. major after several culture passages compared with wild parasites. The title sound novel and would represent valuable information on virulence-related genes that can be finally translated into the clinical settings.

The utilized methods have been appropriately applied towards the manuscript goal. There is only some typographical and stylistic errors in English writing in the manuscript, mostly in introduction and discussion parts, which should be obviated.

In total, the manuscript has been carefully devised and performed, and will be of great interest to the readers of PLoS One journal.

Response: with thanks; some sections were improved as highlighted. 

Reviewer #2: This manuscript presents an interesting concept and is well designed. There are some comments that must be addressed:

-Please write the abstract in a structural form. See the journal guidelines.

-Line 41-42: Please write the full form of gp63, HSP70, LACK, SHERP, CPB in the abstract first, and then write its abbreviation.

-Line 51: Please correct the phrase "Out results showed a significant increased gene..."

-Line 52-54: Please state any significant differences in the abstract text statistically and write p-value.

- According to the sentence of line 99. Explain the reason for choosing the Stationary phase compared whit the logarithmic phase.

-Please mention the reason for choosing seven genes among different virulence factors in the introduction section.

-In line 125, please write Ph. papatasi in italic form and consider in other parts of the text.

-In line 182: write the type of Real Time PCR system and Master Mix in detail.

- In line 185, it refers to the (Table), while there is no table in the text of the manuscript. Please check.

-Please mention the limitations and recommendations in the last paragraph of the discussion.

Response: with thanks for your above constructive comments; all of the comments were obviated in the manuscript text, as highlighted in yellow.

---

## [Decision Letter · Decision Letter 1]

28 Mar 2023

Evaluation of Expression variations in virulence-related genes of Leishmania major after several culture passages compared with Phlebotomus papatasi isolated promastigotes

PONE-D-22-35287R1

Dear Dr. Abbasali Eskandarian,

We’re pleased to inform you that your manuscript has been judged scientifically suitable for publication and will be formally accepted for publication once it meets all outstanding technical requirements.

Kind regards,

Alireza Badirzadeh

Academic Editor

PLOS ONE

Additional Editor Comments (optional):

Reviewers' comments:

Reviewer's Responses to Questions

**Comments to the Author**

1. If the authors have adequately addressed your comments raised in a previous round of review and you feel that this manuscript is now acceptable for publication, you may indicate that here to bypass the “Comments to the Author” section, enter your conflict of interest statement in the “Confidential to Editor” section, and submit your "Accept" recommendation.

Reviewer #2: All comments have been addressed

2. Is the manuscript technically sound, and do the data support the conclusions?

Reviewer #2: Yes

3. Has the statistical analysis been performed appropriately and rigorously? 

Reviewer #2: Yes

4. Have the authors made all data underlying the findings in their manuscript fully available?

Reviewer #2: Yes

5. Is the manuscript presented in an intelligible fashion and written in standard English?

Reviewer #2: Yes

6. Review Comments to the Author

Reviewer #2: Line 53 in abstract section, "Out results" correct to "our results"

Line 103 in introduction section, "amastigotes assists" be corrected to "amastigotes assist"

7. PLOS authors have the option to publish the peer review history of their article (what does this mean?). If published, this will include your full peer review and any attached files.

Reviewer #2: No

---

## [Editor Report · Acceptance letter]

3 Apr 2023

PONE-D-22-35287R1 

Evaluation of expression variations in virulence-related genes of *Leishmania major* after several culture passages compared with *Phlebotomus papatasi* isolated promastigotes 

Dear Dr. Eskandarian:

I'm pleased to inform you that your manuscript has been deemed suitable for publication in PLOS ONE. Congratulations! Your manuscript is now with our production department. 

Kind regards, 

on behalf of

Dr. Alireza Badirzadeh 

Academic Editor

PLOS ONE